# Beneficial Effect of Exogenously Applied Calcium Pyruvate in Alleviating Water Deficit in Sugarcane as Assessed by Chlorophyll a Fluorescence Technique

**DOI:** 10.3390/plants13030434

**Published:** 2024-02-01

**Authors:** Mirandy dos Santos Dias, Francisco de Assis da Silva, Pedro Dantas Fernandes, Carlos Henrique de Azevedo Farias, Robson Felipe de Lima, Maria de Fátima Caetano da Silva, Vitória Régia do Nascimento Lima, Andrezza Maia de Lima, Cassiano Nogueira de Lacerda, Lígia Sampaio Reis, Weslley Bruno Belo de Souza, André Alisson Rodrigues da Silva, Thiago Filipe de Lima Arruda

**Affiliations:** 1Unidade Acadêmica de Engenharia Agrícola—UAEA, Centro de Tecnologia e Recursos Naturais—CTRN, Universidade Federal de Campina Grande–UFCG, Campus Campina Grande, Campina Grande 58428-830, PB, Brazil; agrofdsilva@gmail.com (F.d.A.d.S.); pedrodantasfernandes@gmail.com (P.D.F.); cahique.proojet@gmail.com (C.H.d.A.F.); robson.felipe@estudante.ufcg.edu.br (R.F.d.L.); maria.caetano@estudante.ufcg.edu.br (M.d.F.C.d.S.); regia7665@gmail.com (V.R.d.N.L.); andrezzamaia2010@hotmail.com (A.M.d.L.); cassianonogueiraagro@gmail.com (C.N.d.L.); weslley.bruno@estudante.ufcg.edu.br (W.B.B.d.S.); andre.alisson@estudante.ufcg.edu.br (A.A.R.d.S.); thiago.filipe@estudante.ufcg.edu.br (T.F.d.L.A.); 2Campus de Engenharias e Ciências Agrárias—CECA, Universidade Federal de Alagoas—UFAL, Rio Largo 57100-000, AL, Brazil; lavenere_reis@hotmail.com

**Keywords:** *Saccharum officinarum* L., water scarcity, water deficit mitigation

## Abstract

The growing demand for food production has led to an increase in agricultural areas, including many with low and irregular rainfall, stressing the importance of studies aimed at mitigating the harmful effects of water stress. From this perspective, the objective of this study was to evaluate calcium pyruvate as an attenuator of water deficit on chlorophyll *a* fluorescence of five sugarcane genotypes. The experiment was conducted in a plant nursery where three management strategies (E1—full irrigation, E2—water deficit with the application of 30 mM calcium pyruvate, and E3—water deficit without the application of calcium pyruvate) and five sugarcane genotypes (RB863129, RB92579, RB962962, RB021754, and RB041443) were tested, distributed in randomized blocks, in a 3 × 5 factorial design with three replications. There is dissimilarity in the fluorescence parameters and photosynthetic pigments of the RB863129 genotype in relation to those of the RB041443, RB96262, RB021754, and RB92579 genotypes. Foliar application of calcium pyruvate alleviates the effects of water deficit on the fluorescence parameters of chlorophyll *a* and photosynthetic pigments in sugarcane, without interaction with the genotypes. However, subsequent validation tests will be necessary to test and validate the adoption of this technology under field conditions.

## 1. Introduction

The growing demand for food production has led to an increase in the number of agricultural areas, including the occupation of spaces that have never been farmed before. Sugarcane (*Saccharum officinarum* L.) is a crop that has been widely cultivated in these areas for many years, showing global importance and being used to produce sugar, biofuels, and energy [1]. However, due to its long growth cycle, sugarcane requires an abundant rainfall distribution (1850 to 2500 mm year^−1^), showing greater sensitivity to water deficit during tillering and stem elongation [2,3].

For all crops, global warming and climate change increase the frequency and intensity of abiotic stress factors on a worldwide level, especially water stress [4,5]. In such conditions, the structure of the plant chloroplast is affected by reductions in the chlorophyll content caused by the photooxidation of photosynthetic pigments [6]. As a consequence, a smaller proportion of the incident energy is used to produce adenosine triphosphate (ATP) and nicotinamide adenine dinucleotide phosphate (NADPH), resulting in photoinhibition [7,8], negatively affecting the photosynthetic capacity of plants.

The photosynthetic efficiency of plants is directly related to several factors, including the chlorophyll and carotenoid content, whose leaf values indicate the damage that certain stresses can cause to the photosynthetic apparatus [9]. Furthermore, the evaluation of chlorophyll *a* fluorescence is essential in analyzing photochemical efficiency and damage to the photosynthetic system, providing important information about the inhibition or reduction in electron transfer between photosystems [10,11].

Although the negative effects of water stress on fluorescence emission and the photosynthetic pigment content of sugarcane plants are well reported in the literature [12,13,14,15], there are reports that organic substances can mitigate the harmful effects of drought by increasing the ability of plants to withstand such deleterious conditions, e.g., studies of pyruvate application carried out with peanuts [16] and cotton [17,18]. However, research focusing on the photosynthetic apparatus of sugarcane plants has not yet been carried out.

In the metabolism of C4 plants, pyruvate is responsible for the regeneration of phosphoenolpyruvate, an HCO^3−^ acceptor in mesophyll cells [19]. Furthermore, in the Krebs Cycle metabolism, this substance plays a vital role in converting glucose into energy, in a process in which glucose is broken down into two pyruvic acid molecules that, in the Krebs Cycle, are transformed into energy (ATP) [19]. In this context, it becomes relevant to test technologies that aim to alleviate the harmful effects of water deficit on the photosynthetic apparatus of sugarcane. From this perspective, this study aimed to evaluate calcium pyruvate as an attenuator of water deficit on chlorophyll *a* fluorescence in five sugarcane genotypes.

## 2. Results and Discussion

According to the F-test (Table 1), the initial fluorescence (Fo), the maximum fluorescence (Fm), the variable fluorescence (Fv), the maximum quantum efficiency of PSII (Fv/Fm), the photochemical efficiency (Fv/Fo), the basal quantum efficiency of the non-photochemical process (Fo/Fm), the initial fluorescence before the saturation pulse (F′), the maximum fluorescence after adaptation to saturating light (Fm′), the electron transport rate (ETR), the quantum efficiency of PSII (Y), the Stern–Volmer non-photochemical quenching (NPQ), the complete non-photochemical quenching of chlorophyll fluorescence (QCN), the quantum yield of non-regulated photochemical quenching (YNO), the quantum yield of regulated photochemical quenching (YNPQ), chlorophyll *a* (Chl *a*), chlorophyll *b* (Chl *b*), and carotenoids (Car) were significantly influenced by management strategies (E) (*p* < 0.01). For the genotypes, there was a significant effect on most of the fluorescence parameters studied, with the exception of Fv/Fm, Fv/Fo, Fo/Fm, Y, and YNPQ. The interaction between the management strategies (E) and genotypes (G) was not significant (Table 1).

An increase in Fo was observed (Figure 1A), reaching a mean value of 296.7 electrons quantum^−1^ in irrigated plants; this is significantly lower than the value recorded in E2 (415.8 electrons quantum^−1^) and E3 (473.4 electrons quantum^−1^), corresponding to an increase of 40.14% and 59.56%, respectively. As a result, there was a reduction of 178.4 quantum^−1^ electrons in the maximum fluorescence (−13.47%) and 355.1 electrons quantum^−1^ in the variable fluorescence (−34.55%), when related to the same treatments (Figure 1B,C). The increase in Fo and the reduction in Fv and Fm significantly affected the Fv/Fm and Fv/Fo ratios, with reductions of 15.38% and 24.60% (Fv/Fm) and 44.03% and 59.01% (Fv/Fo), respectively, for E2 and E3 in relation to E1, indicating that the photosynthetic apparatus was compromised (Figure 1D,E). Furthermore, there was an increase in the Fo/Fm ratio of 54.54% and 86.66% when related to the same treatments (Figure 1F). However, when plants that were subjected to water deficit received pyruvate application (E2), there was a reduction in Fo (10.05%) and the Fo/Fm ratio (17.07%) and, consequently, an increase in Fm (7.08%), Fv (20.63%), and the Fv/Fm ratio (10.61%), which was statistically different to plants that did not receive calcium pyruvate (E3).

The Fo increase in plants under water deficit may be the result of damage to the PSII reaction center or a reduced capacity to transfer the excitation energy from the antenna to the reaction center [20,21]. According to Rosseau et al. [22], Fo represents the status of the plant when the PSII reaction center is oxidized. An increase in this parameter directly reflects on Fm and Fv, which affects the flow of electrons between photosystems, thus decreasing the plant’s ability to transfer energy to the formation of NADPH and ATP [10,21].

The maximum quantum efficiency of PSII (Fv/Fm ratio) expresses the capture efficiency of the excitation energy by the open reaction centers of PSII [23,24]. Under normal conditions, the Fv/Fm ratio can vary from 0.75 to 0.85, with drops in this ratio indicating structural damage to the thylakoids, affecting photochemical efficiency, CO_2_ assimilation, and, above all, electron transport [23,25]. Values below 0.75 quantum^−1^ electrons indicate that damage is occurring to the plant’s photosynthetic system. Reductions in the Fv/Fm ratio under water deficit conditions were recorded by Silva et al. [26] when studying the physiological parameters of four sugarcane genotypes (RB72454, RB72910, RB92579, and RB867515) under greenhouse conditions, with the authors finding a mean value of 0.71 electrons quantum^−1^ for the Fv/Fm ratio. 

The lowest Fv/Fo ratio value in the E3 management strategy occurred due to a reduction in the variable fluorescence (Fv) or an increase in the initial fluorescence (Fo), indicating a change in the rate of electron transport from PSII to the primary electron acceptors [27]. Furthermore, the increase in the Fo/Fm ratio and Fo associated with the reduction in the Fv/Fm and Fv/Fo ratios suggests the occurrence of photoinhibition in plants cultivated under the E3 management strategy. According to Silva et al. [24], photoinhibition causes a slow reduction in photosynthesis, indicating that the plant was subjected to a stressful environment.

Therefore, maintaining high Fv/Fm and Fv/Fo ratios and a low Fo/Fm ratio improves the efficiency of radiation use and facilitates CO_2_ assimilation [28]. In this context, although the plants that received pyruvate application obtained only 0.66 electrons quantum^−1^, not reaching the ideal values of Fv/Fm (0.75 to 0.84 electrons quantum^−1^), the supplementation of this compound improved the light absorption system of PSII compared with plants that did not receive this product (0.59 electrons quantum^−1^), suggesting that pyruvate alleviated the deleterious effects of water deficit.

When studying the fluorescence parameters, an increase of 42.48% was observed in the F′ parameter in sugarcane plants under water deficit and without pyruvate application (E3) in relation to treatment E1, and 14.86% in relation to treatment E2 (Figure 2A). In the Fm′, ETR, and Y parameters in plants subjected to treatment E3, the reductions were 30.44%, 69.15%, and 46.97%, respectively, in relation to treatment E1, and 20.26%, 46.23%, and 32.69%, respectively, in relation to treatment E2 (Figure 2B–D).

These results indicate that, without the application of calcium pyruvate (E3), water deficit stress negatively influenced all the chlorophyll fluorescence parameters. These results corroborate those obtained by Souza et al. [13], Leanasawat et al. [15], and Verma et al. [29] in sugarcane under drought conditions, which recorded reductions in fluorescence emission levels.

In addition to the Fv/Fm ratio, the ETR is also considered an important parameter to determine the efficiency of the photosynthetic apparatus [20]. In this research, a negative effect of water deficit on the electron transport rate was observed (Figure 2C), notably in plants that did not receive the foliar application of calcium pyruvate, indicating a reduced capacity for electron transport and compromised production of ATP and NADPH during the continuity of the photosynthetic process.

Through the analysis of the quenching coefficients, it was observed that water deficit during tillering and stalk elongation in sugarcane affected the photosynthetic system, with a significant increase in the NPQ (39.48%), QCN (5.06%), YNO (67.65%), and YNPQ (118.51%) parameters compared with management strategy E1 (Figure 3A–D). However, plants subjected to strategy E2 showed significant reductions (*p* ≤ 0.05) in the NPQ (19.74%), QCN (2.41%), YNO (18.42%), and YNPQ (33.89%) parameters compared with plants subjected to water deficit without calcium pyruvate (Figure 3A–D).

Chlorophylls are excited when plants receive sunlight, making these pigments highly reactive. If there is no attenuation, they will become an element that generates oxidative stress [30]. This occurs because chlorophylls do not stop absorbing light; therefore, all excess light received must be dissipated. If this dissipation is carried out in the form of heat (NPQ), there is an increase in the QCN and YNO parameters, consequently increasing the YNPQ parameter, which corroborates the results found in this study, as the plants subjected to irrigation strategy E3 had reduced Fv/Fm ratio, Y, and ETR parameters.

Such reductions resulted in increased values of the Stern–Volmer non-photochemical quenching (NPQ), complete non-photochemical quenching of chlorophyll fluorescence (QCN), and quantum yield of non-regulated photochemical quenching (YNO) parameters, possibly triggering oxidative processes in the PSII of plants. In this situation, plants need to dissipate the excess absorbed light energy as heat since it exceeds their normal use for driving photosynthesis and electron transfer, thus increasing the quantum yield of regulated photochemical quenching (YNPQ) parameter. However, it appears that foliar supplementation with calcium pyruvate in plants under water deficit (E2) significantly improved all the studied chlorophyll fluorescence parameters.

Photosynthetic pigments play a fundamental role in the growth and development of plants, being responsible for the transmission of light energy for the production of photoassimilates [31]. In the present study, the levels of chlorophyll *a* (Chl *a*), chlorophyll *b* (Chl *b*), and carotenoids in sugarcane leaves were significantly reduced when plants were cultivated under water deficit in the tillering and stalk elongation phases (E2 and E3) compared with plants under full irrigation (E1) (Figure 4). However, the plants that received calcium pyruvate had increased levels of chlorophyll *a* (Figure 4A), *b* (Figure 4B), and carotenoids (Figure 4C), by 55.40%, 43.80%, and 55.70%, respectively, compared with the treatment without calcium pyruvate under water deficit (E3), expressing greater tolerance to this stress condition through the maintenance of photosynthetic pigments. 

The degradation in the levels of chlorophyll *a*, chlorophyll *b*, and carotenoids may be due to the reduced biosynthesis of chlorophyll resulting from photochemical disturbances caused by excess light in the PSII reaction centers [32]. Carotenoids are non-enzymatic antioxidants that target the excessive accumulation of reactive oxygen species [33]. Possibly, the low carotenoid content contributed to reducing the photosynthetic activity during the period of water deficit and increasing photodegradation, which resulted in a low Fv/Fm ratio. Souza et al. [13] also found reductions in fluorescence emission levels and photosynthetic pigment content in genotypes RB855536 and RB93509 that were subjected to water restriction during early growth. 

In addition to sugarcane, there are also reports in the literature about the degradation of photosynthetic pigments in several crops under water deficit, e.g., wheat [34], sorghum [35], and maize [36]. Furthermore, although research focuses only on the harmful effects of water deficit, there are reports that organic substances can reduce the harmful effects of drought on plants. For example, Verma et al. [29] found that the foliar application of silicon, in addition to reducing the harmful effects of water deficit in sugarcane, also improves the plant’s antioxidant defense system, by favoring the synthesis of photosynthetic pigments and the maximum quantum efficiency of photosystem II. In another study, Maia Júnior et al. [37] observed that, in sugarcane plants, the foliar application of glycine betaine mitigates the harmful effects of water deficit on the PSII photochemical apparatus. 

In addition to these studies, Dias et al. [38] found that the effects of water deficit on the tillering and stalk elongation phases in sugarcane are alleviated by the exogenous application of 30 mM calcium pyruvate.

### 2.1. Multivariate Data Analysis

To corroborate the results presented in the univariate analysis, a multivariate analysis of the data was carried out using a cluster analysis and a principal component analysis.

#### 2.1.1. Cluster Analysis

In order to verify the relationship between the genotypes studied, a cluster analysis was carried out by applying the Euclidean Distance (ED) as a measure of dissimilarity. As a subjective criterion for visual inspection, a cutoff was established between 4.0 and 5.0. For this analysis, only the variables with a significant difference between the genotypes were used, as observed in the F-test (Table 1). Three groups were formed: group 1 was formed by genotypes RB041443 and RB962962, group 2 by RB021754 and RB92579, and group 3 by RB863129 (Figure 5). These groups were characterized by greater homogeneity between the genotypes of each group and greater heterogeneity between the groups in relation to the parameters analyzed.

#### 2.1.2. Principal Component Analysis

The first two principal components explained 91.66% of the variance contained in the original variables. Furthermore, factor loadings with an absolute value greater than 0.60 were considered relevant. The first principal component (PC1) contributed 79.65% of the total variance, and the second component (PC2) contributed 12.01% of the remaining variance (Table 2).

The two-dimensional projections of the combinations of the sugarcane genotypes and irrigation management strategies on the first two PCs are illustrated in Figure 6A,B. The first principal component was formed from the combination of 15 physiological parameters, which was divided into two groups: group 1, formed by the parameters Fo, Fo/Fm ratio, F′, YNPQ, and YNO; and group 2, formed by the parameters Fm, Fv, Fv/Fm ratio, Fv/Fo ratio, Fm′, Y, ETR, Chl *a*, Chl *b*, and carotenoids. The groups formed in this component presented variations in opposite directions, i.e., as the parameters of group 1 increased, the parameters of group 2 decreased the physiological activity of the sugarcane genotypes. These changes were more significant in plants subjected to the E3 irrigation management strategy. However, when receiving calcium pyruvate application (E2), reductions were observed in the Fo, Fo/Fm ratio, F′, YNPQ, and YNO parameters and increases in were observed in the Fm, Fv, Fv/Fm ratio, Fv/Fo ratio, Fm′ Y, ETR, Chl *a*, Chl *b*, and carotenoids parameters in all genotypes in relation to the pyruvate-free strategy (E3). The second component is represented by the non-photochemical parameters, QCN and NPQ, which correlated proportionally.

Based on the physiological parameters of sugarcane, the highest values of the Fo, Fo/Fm ratio, and F′ parameters were observed when plants were subjected to strategy E3, influencing the increase of the photochemical quenching parameters QCN, NPQ, YNPQ, and YNO. This increase suggests that there was damage to the plant’s photosynthetic apparatus, since the high initial fluorescence indicates the dissipation of energy lost by the plant. The damage caused to the plant’s photosynthetic apparatus contributed to the reduction in the Fm, Fv, Fv/Fm ratio, Fm′, Y, Chl *a*, Chl *b*, and carotenoids parameters.

## 3. Materials and Methods

### 3.1. Location of This Experiment

This experiment was conducted in a plant nursery belonging to the Agricultural Engineering Academic Unit of the Federal University of Campina Grande (UAEA-UFCG), located in the city of Campina Grande, PB, at the geographic coordinates 7°15′18″ S, 35 °52′28″ W, at a mean elevation of 550 m a.s.l (meters above sea level).

### 3.2. Treatments and Experimental Design

The treatments were obtained from three management strategies (E1—full irrigation throughout the crop cycle, E2—water deficit with the application of 30 mM calcium pyruvate, and E3—water deficit without the application of calcium pyruvate) and five commercial sugarcane genotypes (G1—RB863129, G2—RB92579, G3—RB962962, G4—RB021754, and G5—RB041443), distributed in randomized blocks in a 3 × 5 factorial design with three replications.

The pyruvate concentration (30 mM) was established based on a study developed by Shen et al. [39] with *Arabidopsis thaliana*, in which the authors used the exogenous application of pyruvate. Adjustments were made to the pyruvate concentration since the original research was carried out with a sample of incubated leaves from an uncultivated species (Arabidopsis). The adjustment was also based on the study developed by Barbosa et al. [16], who used 50 mM pyruvate in peanut plants under drought stress. The calcium pyruvate used in this research was chosen because it is considered a low-cost product and is easily found commercially, when compared with the other sources of pyruvate.

### 3.3. Description of the Experiment

Plastic containers with a capacity of 45 L adapted as drainage lysimeters were used in the experiment, receiving a 2.0 cm layer of crushed stone and a non-woven geotextile fabric (Bidim OP 30) at the bottom, and distributed in a 1.0 × 1.5 m spatial arrangement. Two 10.0-mm wide hoses were connected to each lysimeter, which were coupled to two containers with a volumetric capacity of 2.0 L to collect the drained water (Figure 7).

The pots were filled with soil that was classified as Regolithic Neosol (Entisol) with a sandy loam texture. The soil came from the municipality of Lagoa Seca, Paraíba, Brazil, and its physical and chemical attributes were determined according to the methodology described by Teixeira et al. [40] (Table 3).

The soil moisture values at the tensions of −10, −33, −100, −500, −1000, and −1500 kPa were used to adjust the soil water retention curve (Figure 8). In order to obtain the adjustment parameters, the volumetric moisture values (θ) corresponding to the applied matrix potentials (Ψm) were modeled using the RETC v.6 software and the non-linear model proposed by Van Genuchten [41].

After harvest (the first production of culms after planting), the second cultivation cycle began. From this point, the pots were irrigated regularly, close to the level corresponding to field capacity, until the moment of implementation of the treatments.

Fertilization with nitrogen, potassium, and phosphorus was applied weekly via irrigation water, with a total of 47.67 g of urea (45% N), 36.28 g of monoammonium phosphate (51% P_2_O_5_, 11% N), and 71.25 g of potassium chloride (60% K_2_O). The micronutrients were applied at 15-day intervals to avoid nutritional deficiency by applying 1.0 g L^−1^ of Quimifol^®^ (composition: Mg (5.0%); S (11.0%); B (3.5%); Cu (0.10%); Fe (0.20%); Mn (1.0%); Mo (0.10%); and Zn (6.0%)) with the aid of a knapsack sprayer.

Calcium pyruvate was purchased from Natusvita^®^. The solution was obtained by dissolving calcium pyruvate in distilled water minutes before spraying and applied with the aid of a Jacto XP knapsack sprayer with a capacity of 12 L, a working pressure (maximum) of 88 psi (6 bar), and a JD 12P Nozzle. Spraying was carried out at 5:00 p.m. on all the leaves of the plant. Between 100 and 200 mL of the solution was applied per experimental unit. Table 4 describes the period of the application of treatments.

For better adhesion and absorption, an adhesive spreader was added to the solutions during spraying. Furthermore, to avoid drift caused by the wind during spraying, the plants in each pot were protected with plastic, and the soil was covered with an impermeable mantle to prevent runoff to the soil surface (Figure 9).

Irrigation was carried out daily, at 5:00 p.m., by applying the water volume corresponding to the demand of the plant undergoing each treatment. The volume applied to each pot, per irrigation event, was estimated individually using the water balance, according to Equation (1). A 20% leaching fraction was used monthly to remove excess salts from the ground [42].
(1)VI=Va−Vd 
where:VI—the water volume to be used in the irrigation event (mL);Va—the volume applied in the previous irrigation event (mL); andVd—the volume drained, quantified on the next morning (mL).

At 64 and 211 days after regrowth (DAR), soil samples were taken using a mini-auger, and the soil moisture content was determined using the standard oven method. Then, the soil samples were put in aluminum cans and weighed to obtain the wet mass. After this step, they were oven-dried at 105 °C for 72 h to obtain the dry mass, from which the soil moisture content was determined on a gravimetric basis [40], which is related to the soil matric potential (Table 5).

### 3.4. Variables Analyzed

At the end of the water-deficit period (211 DAR), the initial fluorescence (Fo), the maximum fluorescence (Fm), and the variable fluorescence (Fv) were measured using a modulated pulse fluorometer: model OS5p from Opti-Science ((Hudson, NH, USA) Saturation flash intensity: 11,250 μmol and saturating pulse width: 0.8 s). From these results, the maximum quantum efficiency of photosystem II (Fv/Fm), the photochemical efficiency (Fv/Fo), and the basal quantum efficiency of the non-photochemical process (Fo/Fm) were determined. The evaluations were carried out on leaves with blades that were adapted to the dark for 30 min, using a clip to ensure that all the primary acceptors were fully oxidized [43].

After the fluorescence evaluations with adaptation to the dark, the evaluations were carried out under lighting conditions using the ‘Yield’ protocol (Saturation flash intensity: 11,250 μmol, saturating pulse width: 0.8 s, and default PAR: 120 μE) to determine the initial fluorescence before the saturation pulse (F′), the maximum fluorescence after adaptation to saturating light (Fm′), the electron transport rate (ETR), and the quantum efficiency of photosystem II (Y). From these results, the Stern–Volmer non-photochemical quenching coefficient (NPQ), the complete non-photochemical quenching coefficient of chlorophyll fluorescence (QCN), the quantum yield of regulated photochemical quenching (YNPQ), and the quantum yield of non-regulated photochemical quenching (YNO) were determined [44,45].

The content of the photosynthetic pigments (chlorophyll *a*, chlorophyll *b*, and carotenoids) was determined at 211 DAR, according to Arnon [46] and Lichtenthäler [47], by removing a 6-mm disc from the +2 leaf of each treatment. Each sample received 6.0 mL of 80% acetone P.A. Subsequently, the supernatants containing the extracted pigments were collected, and absorbance readings were taken on a spectrophotometer (model UV/VIS-UV1720, AKSON^®^, São Leopoldo, RS, Brazil) at the absorbance wavelengths (ABS) of 470, 645, and 663 nm, calculated using Equations (2)–(4), with the values expressed in micrograms of pigment per gram of fresh mass (μg g^−1^ FW).
(2)$$\mathrm{Clh}\;a=\frac{\left(\left(12.7\times {\mathrm{ABS}}_{663})-(2.79\times {\mathrm{ABS}}_{647}\right)\right)\times V}{\mathrm{FW}}$$
(3)$$\mathrm{Clh}\;b=\frac{\left(\left(22.9\times {\mathrm{ABS}}_{647})-(5.10\times {\mathrm{ABS}}_{663}\right)\right)\times V}{\mathrm{FW}}$$
(4)$$\mathrm{Car}=\frac{\left(\left(\left(1000\times {\mathrm{ABS}}_{470}\right)-\left(1.82\times \mathrm{Clh}\;a\right)-\left(85.02\times \mathrm{Clh}\;b\right)\right)/198\right)\times V}{\mathrm{FW}}$$
where:ABS_470_, ABS_663_, and ABS_647_—the absorbances at 480, 663, and 645 nm, respectively;V—the volume of 80% acetone used in extraction (mL); andFW—the fresh matter (g).

At the end of the analyses in all treatments, soil moisture was restored to close to the level corresponding to field capacity.

### 3.5. Statistical Analysis

Prior to the analysis of variance, the data were subjected to the normality test (Shapiro–Wilk). The analysis of variance and the F-test were carried out with the positive results obtained from the previous tests. Then, the test of means was applied using the Tukey test (*p* ≤ 0.05), using the software Sisvar 5.8 [48]. A multivariate analysis using a principal component analysis (PCA) was also employed; to this end, the data were normalized to a zero mean (= 0.0), unit variance (σ^2^) = 1.0, eigenvalues (λ) > 1.0, and a total variance (σ^2^) > 10% [49]. Only variables with a Pearson correlation coefficient above 0.6 were kept in the composition of each principal component (PC) [50]. The analyses were processed using the software Statistica 7.0 [51].

## 4. Conclusions

There is a dissimilarity in the fluorescence parameters and photosynthetic pigments of genotype RB863129 in relation to those of genotypes RB041443, RB96262, RB021754, and RB92579.

Foliar application of calcium pyruvate alleviates the deleterious effects of water deficit in sugarcane on the fluorescence parameters of chlorophyll *a* and photosynthetic pigments, without interaction with the genotypes. However, subsequent validation tests will be necessary to test and validate the adoption of this technology under field conditions.

## Figures and Tables

**Figure 1 plants-13-00434-f001:**
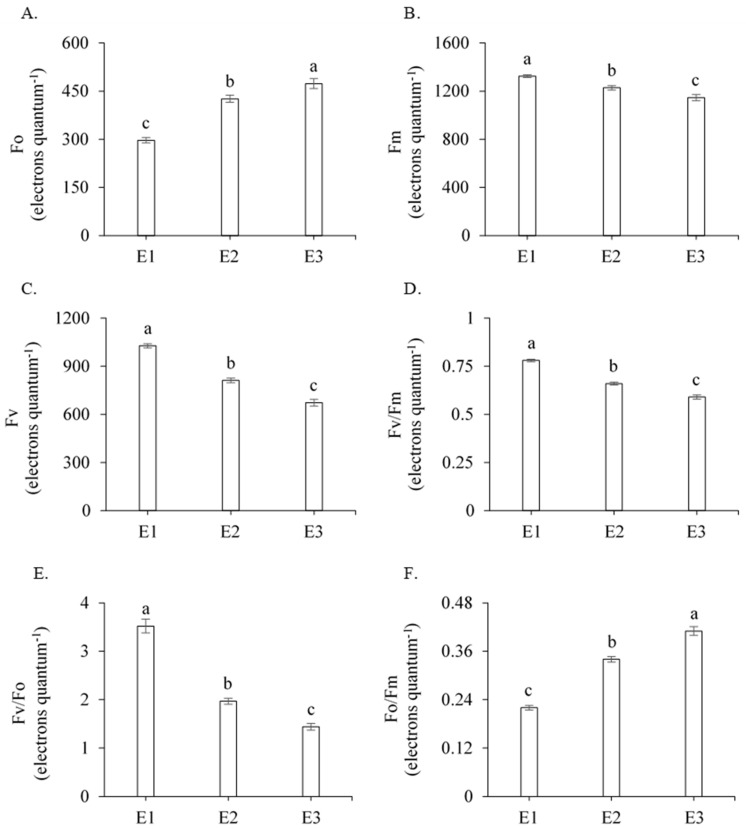
Mean values for initial fluorescence (Fo) (**A**), maximum fluorescence (Fm) (**B**), variable fluorescence (Fv) (**C**), maximum quantum efficiency of PSII (Fv/Fm) (**D**), photochemical efficiency (Fv/Fo) (**E**), and basal quantum efficiency of the non-photochemical process (Fo/Fm) (**F**) of sugarcane as a function of the three management strategies (E), 211 days after regrowth. Different lowercase letters indicate a statistical difference using the Tukey test (*p* ≤ 0.05). Vertical bars represent the standard error of the mean (n = 3). E1—full irrigation, E2—water deficit plus 30 mM of calcium pyruvate, E3—water deficit without calcium pyruvate.

**Figure 2 plants-13-00434-f002:**
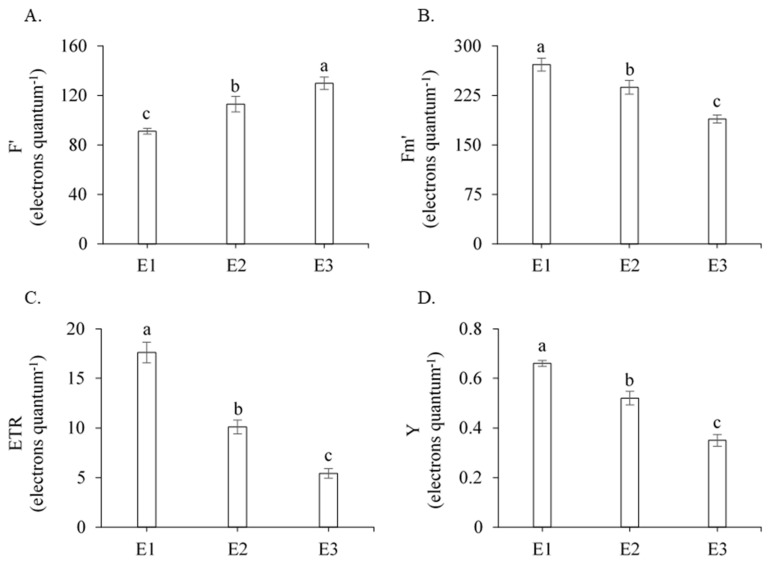
Mean values of initial fluorescence before the saturation pulse (F′) (**A**), maximum fluorescence after adaptation to saturating light (Fm′) (**B**), electron transport rate (ETR) (**C**), and quantum efficiency of PSII (Y) (**D**) of sugarcane as a function of the three management strategies (E), 211 days after regrowth. Different lowercase letters indicate statistical difference using the Tukey test (*p* ≤ 0.05). Vertical bars represent the standard error of the mean (n = 3). E1—full irrigation, E2—water deficit plus 30 mM of calcium pyruvate, E3—water deficit without calcium pyruvate.

**Figure 3 plants-13-00434-f003:**
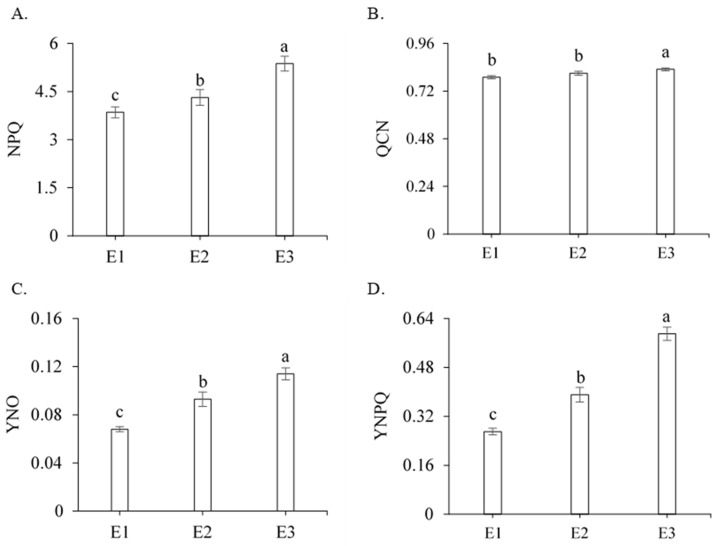
Mean values for Stern–Volmer non-photochemical quenching (NPQ) (**A**), complete non-photochemical quenching of chlorophyll fluorescence (QCN) (**B**), quantum yield of non-regulated photochemical quenching (YNO) (**C**), and quantum yield of regulated photochemical quenching (YNPQ) (**D**) in sugarcane as a function of the three management strategies (E), 211 days after regrowth. Different lowercase letters indicate a statistical difference using the Tukey test (*p* ≤ 0.05). Vertical bars represent the standard error of the mean (n = 3). E1—full irrigation, E2—water deficit plus 30 mM of calcium pyruvate, E3—water deficit without calcium pyruvate.

**Figure 4 plants-13-00434-f004:**
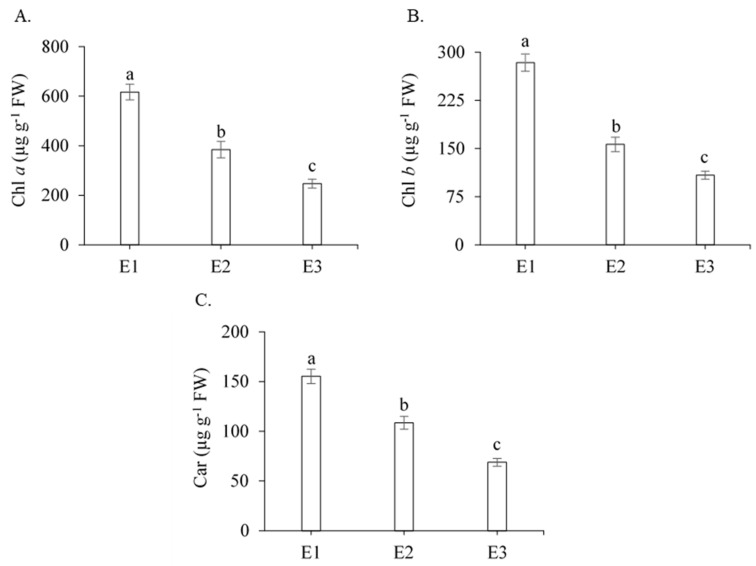
Mean values for chlorophyll *a* (**A**), chlorophyll *b* (**B**), and carotenoids (**C**) of sugarcane according to the three management strategies (E), 211 days after regrowth. Different lowercase letters indicate statistical difference using the Tukey test (*p* ≤ 0.05). Vertical bars represent the standard error of the mean (n = 3). E1—full irrigation, E2—water deficit plus 30 mM of calcium pyruvate, E3—water deficit without calcium pyruvate.

**Figure 5 plants-13-00434-f005:**
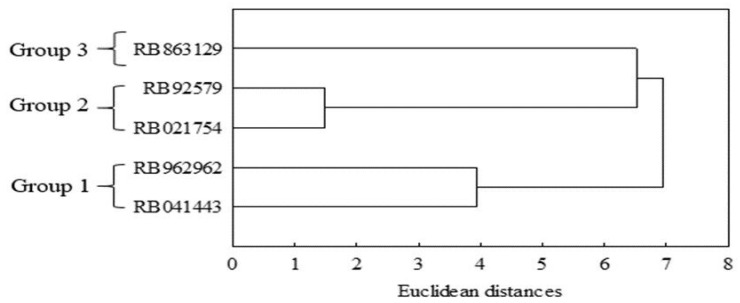
Clustering of sugarcane genotypes assembled with variables according to the isolated effect for genotypes.

**Figure 6 plants-13-00434-f006:**
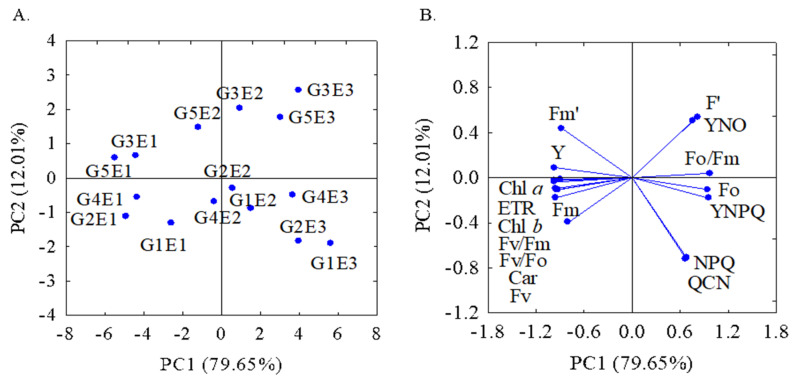
Two-dimensional projection of the principal component scores for the genotypes and treatments (**A**) and the sugarcane variables analyzed (**B**) in the first two principal components (PC1 and PC2). Initial fluorescence (Fo), maximum fluorescence (Fm), variable fluorescence (Fv), maximum quantum efficiency of photosystem II (Fv/Fm), maximum primary efficiency of the photochemical process in PSII (Fv/Fo), basal quantum efficiency of the non-photochemical process in PSII (Fo/Fm), initial fluorescence before the saturation pulse (F′), maximum fluorescence after adaptation to saturating light (Fm′), quantum efficiency of photosystem II (Y), electron transport rate (ETR), complete non-photochemical quenching of chlorophyll fluorescence (QCN), quantum yield of regulated photochemical quenching (YNPQ), quantum yield of non-regulated photochemical quenching (YNO), and content of chlorophyll *a* (Chl *a*), chlorophyll *b* (Chl *b*), and carotenoids (Car) in sugarcane plants grown under water deficit and calcium pyruvate application at 211 DAR. G1—RB863129, G2—RB92579, G3—RB962962, G4—RB021754, and G5—RB041443). E1—full irrigation, E2—water deficit plus 30 mM of calcium pyruvate, E3—water deficit without calcium pyruvate.

**Figure 7 plants-13-00434-f007:**
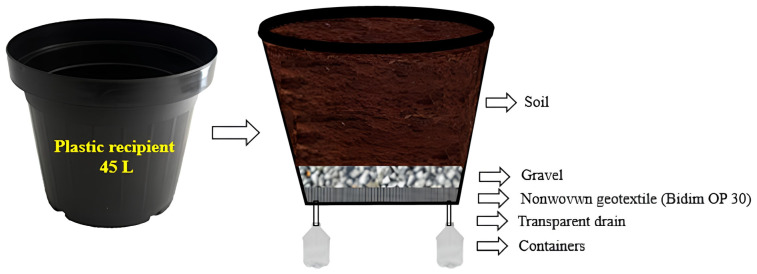
Illustration of the preparation of the drainage lysimeters for soil filling.

**Figure 8 plants-13-00434-f008:**
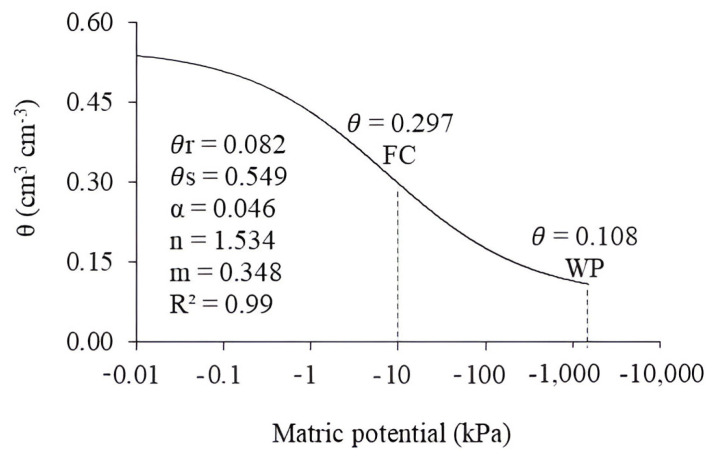
Soil water retention curve according to the van Genuchten model. *θ*r: residual soil moisture; *θ*s: soil moisture at saturation; α, n, and m are the model adjustment parameters; FC: field capacity and WP: wilting point.

**Figure 9 plants-13-00434-f009:**
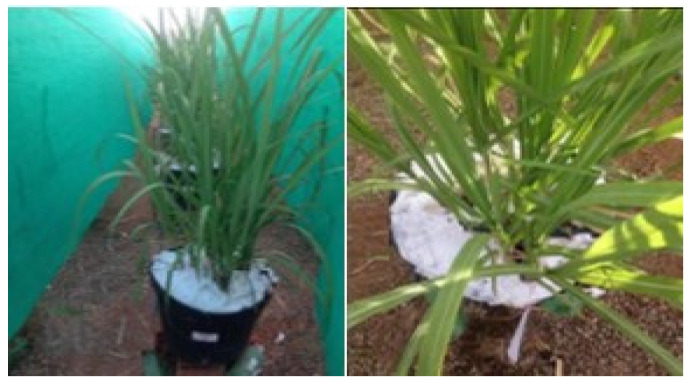
Protection of the sugarcane plants to prevent drift to other plants and runoff to the soil surface during spraying.

**Table 1 plants-13-00434-t001:** F-test for chlorophyll *a* fluorescence and photosynthetic pigments in sugarcane genotypes under management strategies, 211 days after regrowth (DAR).

Sources of Variation	F-Test									
	Fo	Fm	Fv	Fv/Fm	Fv/Fo	Fo/Fm	F′	Fm′	ETR	Y
Strategies (E)	**	**	**	**	**	**	**	**	**	**
Genotype (G)	*	**	*	ns	ns	ns	*	**	**	ns
G × E	ns	ns	ns	ns	ns	ns	ns	ns	ns	ns
Block	*	**	ns	ns	ns	ns	ns	ns	ns	ns
CV (%)	9.85	4.16	6.53	4.36	16.62	9.03	15.90	12.06	9.85	17.98
	NPQ	QCN		YNO		YNPQ	Chl *a*		Chl *b*	Car
Strategies (E)	**	**		**		**	**		**	**
Genotype (G)	**	**		*		ns	**		**	**
G × E	ns	ns		ns		ns	ns		ns	ns
Block	ns	ns		ns		ns	ns		ns	ns
CV (%)	14.96	2.96		18.69		19.92	18.50		17.91	13.83

*, **, significant at 1% and 5%, respectively. ns—not significant. CV (%)—coefficient of variation.

**Table 2 plants-13-00434-t002:** Eigenvalues, percentage of total variance explained, and correlation coefficients (r) between the original variables and the principal components.

						Principal Components			
						PC1		PC2	
Eigenvalues (λ)						13.54		2.02	
Percentage of Total Variance (S^2^%)						79.65		12.01	
PCs	Correlation coefficient (r)								
	Fo	Fm	Fv	Fv/Fm	Fv/Fo	Fo/Fm	F′	Fm′	Y
PC1	0.94 *	−0.80 *	−0.96 *	−0.97 *	−0.96 *	0.97 *	0.76 *	−0.88 *	−0.97 *
PC2	−0.11	−0.40	−0.18	−0.04	−0.10	−0.04	0.50	0.44	0.10
	ETR	QCN	NPQ	YNPQ	YNO	Chl *a*	Chl *b*	Car	
PC1	−0.98 *	0.68	0.69	0.95 *	0.81 *	−0.94 *	−0.90 *	−0.93 *	
PC2	−0.03	−0.72 *	−0.71 *	−0.18	0.54	−0.11	−0.01	−0.11	

* Variables considered in PCA: r = 0.10–0.39 (weak), 0.40–0.69 (moderate), and 0.70–1.00 (strong).

**Table 3 plants-13-00434-t003:** Chemical and physical attributes of the soil used in the experiment.

Chemical Attributes			Physical Attributes	
			Sand	63.48%
pH	6.50	-	Silt	25.14%
P	79.0	mg dm^−3^	Clay	11.38%
K^+^	0.24	cmolc dm^−3^	Soil density	1.13 g cm^−3^
Ca^2+^	9.50	cmolc dm^−3^	Particle density	2.72 g cm^−3^
Na^+^	0.51	cmolc dm^−3^	Porosity	58.45%
Mg^2+^	5.40	cmolc dm^−3^	Sandy loam	
Al^3+^	0.00	cmolc dm^−3^	Matric potential (kPa)	Moisture (%)
H^+^	0.90	cmolc dm^−3^	Natural	0.55
SB	15.65	cmolc dm^−3^	−10	24.86
CEC	16.55	cmolc dm^−3^	−33	17.05
V	94.56	%	−100	12.57
M	0.00	%	−500	9.01
OM	8.10	g dm^−3^	−1000	8.91
			−1500	8.84

pH (H_2_O)—potential of hydrogen; SB—sum of bases; CEC—cation exchange capacity at pH 7.0; Mehlich (P, K, Na); Potassium chloride (KCl) 1N (Ca, Mg, and Al); Calcium acetate at pH 7.0 (H + Al); OM—organic matter; V—base saturation, and M—aluminum saturation.

**Table 4 plants-13-00434-t004:** Application of treatments in sugarcane plants.

	Water Deficit		Applications		
	Period (DAR)	Total (Days)	Calcium Pyruvate	Water	Total
			Period (DAR)		
E1	-	-	-	-	-
E2	24 to 64 and 182 to 211	71	39 to 63 and 192 to 210	-	23
E3	24 to 64 and 182 to 211	71	-	39 to 63 and 192 to 210 *	23

* Plants under water deficit that did not receive calcium pyruvate (E3) were sprayed with distilled water plus an adhesive spreader. DAR—days after regrowth. Applications were carried out at two-day intervals between each application. E1—full irrigation, E2—water deficit plus 30 mM of calcium pyruvate, E3—water deficit without calcium pyruvate.

**Table 5 plants-13-00434-t005:** Soil water moisture and matric potential (Ψm) recorded at the end of the water-deficit period in each treatment.

Genotypes	64 DAR					
	E1		E2		E3	
	Moisture (cm^3^ cm^−3^)	Ψm (kPa)	Moisture (cm^3^ cm^−3^)	Ψm (kPa)	Moisture (cm^3^ cm^−3^)	Ψm (kPa)
RB863129	0.259	−18.6	0.149	−207.9	0.138	−305.8
RB92579	0.240	−25.9	0.143	−254.6	0.145	−237.5
RB962962	0.240	−25.9	0.142	−263.7	0.141	−273.3
RB021754	0.266	−16.5	0.145	−237.5	0.139	−294.1
RB041443	0.244	−24.1	0.149	−207.9	0.148	−214.8
	**211 DAR**					
	**E1**		**E2**		**E3**	
	**Moisture** **(cm^3^ cm^−3^)**	**Ψm (kPa)**	**Moisture** **(cm^3^ cm^−3^)**	**Ψm (kPa)**	**Moisture** **(cm^3^ cm^−3^)**	**Ψm (kPa)**
RB863129	0.242	−25.0	0.126	−504.4	0.135	−342.9
RB92579	0.257	−19.2	0.133	−371.6	0.131	−403.9
RB962962	0.240	−25.9	0.133	−371.6	0.133	−371.6
RB021754	0.234	−28.8	0.127	−481.6	0.129	−440.9
RB041443	0.256	−19.6	0.130	−421.6	0.131	−403.9

Ψm—matric potential. DAR—days after regrowth. E1—full irrigation, E2—water deficit plus 30 mM of calcium pyruvate, E3—water deficit without calcium pyruvate.

## Data Availability

Data are contained within the article.

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
