# Peer review of "Beneficial Effect of Exogenously Applied Calcium Pyruvate in Alleviating Water Deficit in Sugarcane as Assessed by Chlorophyll a Fluorescence Technique"

_plants, 2024, doi:10.3390/plants13030434_

Round 1
Reviewer 1 Report
Comments and Suggestions for Authors
The paper of dos Santos Dias et al. (registered under the ID number; ID# 2815713) unveils the mitigating effects of Calcium pyruvate (C6H6CaO6) on the water deficit in sugarcane using five different genotypes. Authors also investigated the interaction genotype vs treatments to unravel the effect of environment on the phenotype. Based on their findings, authors have shown that under E2 treatment the effect of water deficit on the photosynthetic parameters was less compared to that under E3 treatment. This emphasizes that the Calcium pyruvate somehow relieved the effect of water stress on the photosynthetic apparatus in Sugarcane (mechanism not deciphered), as assessed by the Chlorophyll fluorescence parameters. This work was conducted on a C4 photosynthesis-type species which is known to regenerate phosphoenolpyruvate from Pyruvate. The phosphoenolpyruvate functions as a bicarbonate (HCO3–) acceptor in the mesophyll Cells (MCs) in C4-type photosynthesis. Besides, pyruvate plays other major roles in the Krebs cycle metabolic pathway. The mitigation of water deficit by Calcium pyruvate is, very likely, ascribable to the influence of Calcium pyruvate on the metabolic pathways mediated by phosphoenolpyruvate. This work can be considered for publication, after revision.
Mostly the results are consistent and coherent. So, I support the consideration of this work and its publication after adjusting the following points:
- Title: “Quantum photosynthetic efficiency of chlorophyll a in sugarcane under water deficit and calcium pyruvate application”.
New title: Beneficial effect of exogenously applied calcium pyruvate in alleviating water deficit in sugarcane as assessed (or probed) by chlorophyll a fluorescence technique.
I suggest to use the new title, it is more relevant with the content of your paper.
- Line 38: you mentioned: “.., global warming and climate change are increasing the frequency ….”. English tense problem, please correct and use simple present tense in this situation.
Write as ensuing: global warming and climate change increase the frequency.
- Line 47: photosynthetic system
Correction: please replace “system” by machinery or apparatus.
- Line 55: “…plants to adjust to such conditions,”
Correction: please replace “adjust” by “withstand” (plants to withstand such deleterious conditions)
- For Figs. 1 and 2; Y-axis title should be (Electrons quantum-1) and not (Elétrons quantum-1). No French tongue please.
- Lines 210-211: you wrote this sentence: “As a result, plants needed to dissipate the energy generated by light absorption since it was not being used for photosynthesis”
Please write as follows: In this situation, plants need to dissipate the excess of absorbed light energy as heat since it exceeds their normal use for driving photosynthesis and electrons transfer.
- Line 214: you wrote: “all chlorophyll fluorescence parameters studied”.
Please write as follows: all the studied chlorophyll fluorescence parameters.
- Line 241: you wrote: “during initial growth”
Correction: please replace initial by early
- Line 253: you wrote: “…… the effects of water deficit in the tillering and stalk …”
Please write: …… the effects of …………..on the tillering and stalk …… (on not in)
- Line 335: you wrote: “…… at a mean elevation of 550 m a.s.l.”
Remark: please add between brackets the full text for the abbreviation m. asl (meters above sea level).
- Protocol of Chl a measurement was not clearly illustrated (or described) in Material and methods section, which kind of apparatus was used? Is it the Dual PAM-100 or which machine was employed to achieve your Chl a fluorescence measurements? So please check that, you did not mention the time of light illumination or the saturating pulse and the width. Please verify that and you can refer to the following scientists for help;
* Several papers talked about the chlorophyll fluorescence measurements using saturating pulse method with the PAM-100, such as those of Klughammer and Schreiber, Govindjee.
As you focused your study on Chlorophyll fluorescence parameters, I would like, if possible, to uncover the effect of Calcium pyruvate on the prompt (fast) Chl a fluorescence induction on the so-called OJIP curves. It is a very easy technique to probe the effect of abiotic stresses on the Chl a fluorescence mainly measured on dark-adapted plants for at least one hour. The protocol can be set to just 0.5 or 1 second so, it is very fast, informative and powerful tool. Due, to the long-life cycle of Sugarcane, this experiment can be conducted on an early growth stage only.
* On light of my sentence in the introduction “The mitigation of water deficit by Calcium pyruvate is, very likely, ascribable to the influence of Calcium pyruvate on the metabolic pathways mediated by phosphoenolpyruvate”
Can you please provide a mechanistic model, with two components (normal condition and water deficit), displaying the paramount of Calcium pyruvate in alleviating water deficit in sugarcane, which might be potentially owing to its beneficial effects on the pyruvate metabolic pathways in C4-type photosynthesis?
Generally, the English is ok but it can be improved by checking some sentence structure and tenses. English text can be and should be improved. Especially, in tenses used throughout the manuscript text.
Comments on the Quality of English Language
English can be improved, especially in the tenses use throughout the manuscript text.
Author Response
Dear Editor
The authors are grateful to you and the unanimous Reviewers for the positive and constructive comments and suggestions on our manuscript entitled “Quantum photosynthetic efficiency of chlorophyll a in sugarcane under water deficit and calcium pyruvate application”. The authors would like to inform you that a thorough revision of the manuscript was made, incorporating the suggestions and adopting the text according to the comments. Attached is the revised version of the manuscript. All changes in the text are highlighted in red color.
The authors remain at your disposal for any further information and explanation.
The responses/explanations to the issues raised by the Reviewer 1/Editor are presented below:
REVIEWER 1
- Title: “Quantum photosynthetic efficiency of chlorophyll a in sugarcane under water deficit and calcium pyruvate application”.
Response: Dear reviewer, as suggested, the new title was accepted: Beneficial effect of exogenously applied calcium pyruvate in alleviating water deficit in sugarcane as assessed by chlorophyll a fluorescence technique.
- - Line 38: you mentioned: “.., global warming and climate change are increasing the frequency ….”. English tense problem, please correct and use simple present tense in this situation.
Response: Correction: global warming and climate change increase the frequency.
- Line 47: photosynthetic system
Response: Correction: please replace “system” by apparatus.
- - Line 55: “…plants to adjust to such conditions,”
Response: Correction: please replace “adjust” by “withstand” (plants to withstand such deleterious conditions)
- - For Figs. 1 and 2; Y-axis title should be (Electrons quantum-1) and not (Elétrons quantum-1).
Response: Dear Reviewer, the title Elétrons Quantum-1 from the Y axis has been corrected
- Lines 210-211: you wrote this sentence: “As a result, plants needed to dissipate the energy generated by light absorption since it was not being used for photosynthesis”
Response: Correction: In this situation, plants need to dissipate the excess of absorbed light energy as heat since it exceeds their normal use for driving photosynthesis and electrons transfer.
- Line 214: you wrote: “all chlorophyll fluorescence parameters studied”.
Response: Correction: all the studied chlorophyll fluorescence parameters.
- Line 241: you wrote: “during initial growth”
Response: Correction: during early growth
- Line 253: you wrote: “… the effects of water deficit in the tillering and stalk …”
Response: Correction: … the effects of water deficit on the tillering and stalk…
- Line 335: you wrote: “…… at a mean elevation of 550 m a.s.l.”
Response: Dear Reviewer the full text of the abbreviation m a.s.l has been added in square brackets (meters above sea level).
- Protocol of Chl a measurement was not clearly illustrated (or described) in Material and methods section, which kind of apparatus was used? Is it the Dual PAM-100 or which machine was employed to achieve your Chl a fluorescence measurements? So please check that, you did not mention the time of light illumination or the saturating pulse and the width.
Response: Dear reviewer, as described in the material and methods, chlorophyll a fluorescence measurements were performed using a fluorometer, model OS5p from Opti Science, using a saturating actinic light pulse (> 6000 µmol m-2 s-1).
- * On light of my sentence in the introduction “The mitigation of water deficit by Calcium pyruvate is, very likely, ascribable to the influence of Calcium pyruvate on the metabolic pathways mediated by phosphoenolpyruvate”.
Can you please provide a mechanistic model, with two components (normal condition and water deficit), displaying the paramount of Calcium pyruvate in alleviating water deficit in sugarcane, which might be potentially owing to its beneficial effects on the pyruvate metabolic pathways in C4-type photosynthesis?
Response: Dear reviewer, the two components (normal condition and water deficit) are reasoned in the work. E1 - normal condition and E3 - water deficit. According to the results observed in this research, there was a significant difference between normal conditions and water deficit for all fluorescence parameters. When calcium pyruvate was applied to plants under water deficit (E2), benefits were observed in these parameters in relation to water deficit (E3).
- English can be improved, especially in the tenses use throughout the manuscript text.
Response: Dear reviewer, as requested, an English review was carried out by a specialized company.
Yours sincerely,
Mirandy dos Santos Dias
Corresponding author

Reviewer 2 Report
Comments and Suggestions for Authors
This manuscript focuses on the photosynthetic efficiency of chlorophyll a in sugarcane under water deficit stress and calcium pyruvate application. The authors determined many parameters, but the manuscript needs major revision before it can be considered for publication.
Major Points:
The manuscript is inadequate in that only the effect of calcium pyruvate on some parameters of photosynthesis was determined, and the mechanism of the effect was not investigated.
1) All data without error bars and experiments without biological replication?
2) How do authors determine the optimal concentration of calcium pyruvate? The authors should add a calcium pyruvate gradient experiment. In addition, the authors need to add other pyruvates to verify whether the recovery effect of photosynthesis under water deficit stress is a function of pyruvate or calcium ions.
Minor Points:
1) Abstracts are mainly used to present the main results and conclusions without describing the experimental methodology;
2) Keywords are keywords for the research content and need to be critical and accurate;
Water stress? Water scarcity or flooding? Biotechnology, what does it refer to?
3) In Figures, ‘electrons’ is misspelled as ‘eléctrons’;
4) In Figures, the font size of the title note should be consistent;
5) Line 266, ‘Euclidean Distance (DE)’ should be ‘Euclidean Distance (ED)’;
6) Table 3, Ca+2 should be Ca2+…
7) Some of the Figures in the Materials and Methods section should be placed in the supplementary materials.
Author Response
Dear Editor
The authors are grateful to you and the unanimous Reviewers for the positive and constructive comments and suggestions on our manuscript entitled “Quantum photosynthetic efficiency of chlorophyll a in sugarcane under water deficit and calcium pyruvate application”. The authors would like to inform you that a thorough revision of the manuscript was made, incorporating the suggestions and adopting the text according to the comments. Attached is the revised version of the manuscript. All changes in the text are highlighted in red color.
The authors remain at your disposal for any further information and explanation.
The responses/explanations to the issues raised by the Reviewer 2/Editor are presented below:
REVIEWER 2
- All data without error bars and experiments without biological replication?
Response: Dear reviewer, an error bar has been inserted in all graphs. The experiment consisted of three biological replications, as described in the methodology.
- How do authors determine the optimal concentration of calcium pyruvate? The authors should add a calcium pyruvate gradient experiment. In addition, the authors need to add other pyruvates to verify whether the recovery effect of photosynthesis under water deficit stress is a function of pyruvate or calcium ions.
Response: Dear reviewer, we thank you for the suggestion, however, the best concentration of calcium pyruvate has not yet been determined. This is initial work with the aim of finding out whether pyruvate influences the relief of water deficit. Subsequently, the research group will develop work with concentration gradients. The concentration used in this study was based on works carried out with exogenous application of pyruvic acid.
- Abstracts are mainly used to present the main results and conclusions without describing the experimental methodology.
Response: Dear reviewer, the abstract follows the word limitation rules required by the journal (200 words), briefly describing context, methods (main methods or treatments applied), results (main conclusions of the article) and conclusions.
- Keywords are keywords for the research content and need to be critical and accurate; Water stress? Water scarcity or flooding? Biotechnology, what does it refer to?
Response: Dear reviewer, ‘Water stress’ has been replaced by water scarcity; ‘biotechnology’ to water deficit mitigation.
- In Figures, ‘elétrons’ is misspelled as ‘electrons’;
Response: Dear reviewer, the name elétrons was replaced by electrons
- In Figures, the font size of the title note should be consistent;
Response: Dear reviewer, the font was adjusted in the title of the figures
- Line 266, ‘Euclidean Distance (DE)’ should be ‘Euclidean Distance (ED)’;
Response: Dear reviewer, corrected
- Table 3, Ca+2 should be Ca2+…
Response: Dear reviewer, it has been replaced Ca+2 by Ca2+
7) Some of the Figures in the Materials and Methods section should be placed in the supplementary materials.
Response: Dear reviewer, it was not indicated which figures in the material and methods section could be included as supplementary material.
Yours sincerely,
Mirandy dos Santos Dias
Corresponding author

Reviewer 3 Report
Comments and Suggestions for Authors
This paper focus on physiological items,how to apply in the field?
Author Response
Dear Editor
The authors are grateful to you and the unanimous Reviewers for the positive and constructive comments and suggestions on our manuscript entitled “Quantum photosynthetic efficiency of chlorophyll a in sugarcane under water deficit and calcium pyruvate application”. The authors would like to inform you that a thorough revision of the manuscript was made, incorporating the suggestions and adopting the text according to the comments. Attached is the revised version of the manuscript. All changes in the text are highlighted in red color.
The authors remain at your disposal for any further information and explanation.
The responses/explanations to the issues raised by the Reviewer 3/Editor are presented below:
REVIEWER 3
- This paper focus on physiological items, how to apply in the field?
Response: Dear reviewer, the physiological indices determined in this research can be applied in field conditions, as fluorescence analyzes are performed with the aid of a modulated pulse fluorometer. In field conditions, these indices are essential to verify the water status of plants, consequently, deciding which strategies are appropriate for crop management.
Yours sincerely,
Mirandy dos Santos Dias
Corresponding author

Round 2
Reviewer 1 Report
Comments and Suggestions for Authors
11. Protocol of Chl a measurement was not clearly illustrated (or described) in Material and methods section, which kind of apparatus was used? Is it the Dual PAM-100 or which machine was employed to achieve your Chl a fluorescence measurements? So please check that, you did not mention the time of light illumination or the saturating pulse and the width.
Response: Dear reviewer, as described in the material and methods, chlorophyll a fluorescence measurements were performed using a fluorometer, model OS5p from Opti Science, using a saturating actinic light pulse (> 6000 µmol m-2 s -1 )
* Authors have to add these insights (saturating Pulse, kind of apparatus used and the saturating pulse width) employed to measure Chlorphyll fluorescence induction as an independent subheading in M&M (material and Meth) section. In another term, the machine setup to measure Chl fluorescence need to be shown
Author Response
Dear Editor
The authors are grateful to you and the unanimous Reviewers for the positive and constructive comments and suggestions on our manuscript entitled “Quantum photosynthetic efficiency of chlorophyll a in sugarcane under water deficit and calcium pyruvate application”. The authors would like to inform you that a thorough revision of the manuscript was made, incorporating the suggestions and adopting the text according to the comments. Attached is the revised version of the manuscript. All changes in the text are highlighted in red color.
The authors remain at your disposal for any further information and explanation.
The responses/explanations to the issues raised by the Reviewer 1/Editor are presented below:
REVIEWER 1
- * Authors have to add these insights (Saturating pulse, kind of apparatus used and the saturating pulse width) employed to measure Chlorphyll fluorescence induction as an independent subheading in M&M (material and Meth) section. In another term, the machine setup to measure Chl fluorescence need to be shown.
Response: Dear reviewer, information to measure Chlorphyll fluorescence induction (Saturating pulse, kind of apparatus used and the saturating pulse width) has been were added to the text.
Yours sincerely,
Mirandy dos Santos Dias
Corresponding author

Reviewer 2 Report
Comments and Suggestions for Authors
The manuscript has improved quite a bit from the last one, but still needs further revisions:
Major Point.
The authors mention in the Response to Reviewers that 'This is initial work with the aim of finding out whether pyruvate influences the relief of water deficit. Subsequently, the research group will develop work with concentration gradients.' I agree with this statement, but I think it is acceptable to not be meticulously precise about the concentration used, but there always needs to be a reason why 30mM was chosen instead of 20mM or 40mM? Also, the authors still haven't answered why calcium pyruvate was used and not other pyruvates? If this paper just randomly administered a certain concentration of calcium pyruvate and tested the corresponding physiological indices, I think the study setup itself is imperfect.
Minor Point.
The authors provided error bars this time, but did not tell if the error was ±SD or ±SEM?
Author Response
Dear Editor
The authors are grateful to you and the unanimous Reviewers for the positive and constructive comments and suggestions on our manuscript entitled “Quantum photosynthetic efficiency of chlorophyll a in sugarcane under water deficit and calcium pyruvate application”. The authors would like to inform you that a thorough revision of the manuscript was made, incorporating the suggestions and adopting the text according to the comments. Attached is the revised version of the manuscript. All changes in the text are highlighted in red color.
The authors remain at your disposal for any further information and explanation.
The responses/explanations to the issues raised by the Reviewer 2/Editor are presented below:
REVIEWER 2
- The authors mention in the Response to Reviewers that 'This is initial work with the aim of finding out whether pyruvate influences the relief of water deficit. Subsequently, the research group will develop work with concentration gradients.' I agree with this statement, but I think it is acceptable to not be meticulously precise about the concentration used, but there always needs to be a reason why 30mM was chosen instead of 20mM or 40mM? Also, the authors still haven't answered why calcium pyruvate was used and not other pyruvates? If this paper just randomly administered a certain concentration of calcium pyruvate and tested the corresponding physiological indices, I think the study setup itself is imperfect.
Response: Dear reviewer, the pyruvate concentration (30 mM) was established based on a study developed by Shen et al. [2017] with Arabidopsis thaliana, in which the authors used exogenous application of pyruvate. Adjustments were made to the pyruvate concentration since the original research was carried out with a sample of incubated leaves from an uncultivated species (Arabidopsis). The adjustment was also based on the study developed by Barbosa et al. [2021], who used 50 mM pyruvate in peanut plants under drought stress. The calcium pyruvate used in this research was chosen because it is considered a low-cost product and is easily found commercially, when compared to other sources of pyruvate. This information has been added to the text.
- The authors provided error bars this time, but did not tell if the error was ±SD or ±SEM?
Response: Dear reviewer, bars standard error of the mean were added. This information has been added to the text.
Yours sincerely,
Mirandy dos Santos Dias
Corresponding author

Round 3
Reviewer 2 Report
Comments and Suggestions for Authors
This manuscript could be accepted in current form.